# Detection of Exotic Mosquito Species (Diptera: Culicidae) at International Airports in Europe

**DOI:** 10.3390/ijerph17103450

**Published:** 2020-05-15

**Authors:** Adolfo Ibáñez-Justicia, Nathalie Smitz, Wietse den Hartog, Bart van de Vossenberg, Katrien De Wolf, Isra Deblauwe, Wim Van Bortel, Frans Jacobs, Alexander G. C. Vaux, Jolyon M. Medlock, Arjan Stroo

**Affiliations:** 1Centre for Monitoring of Vectors, Netherlands Food and Consumer Product Safety Authority, Geertjesweg 15, 6706 EA Wageningen, The Netherlands; w.g.s.a.denhartog@nvwa.nl (W.d.H.); F.H.H.Jacobs@nvwa.nl (F.J.); c.j.stroo@nvwa.nl (A.S.); 2Royal Museum for Central Africa (BopCo), Leuvensesteenweg 13–17, 3080 Tervuren, Belgium; nathalie.smitz@africamuseum.be; 3Molecular Biology Group, Netherlands Food and Consumer Product Safety Authority, Geertjesweg 15, 6706 EA Wageningen, The Netherlands; b.t.l.h.vandevossenberg@nvwa.nl; 4Unit of Entomology, Institute of Tropical Medicine, Nationalestraat 155, 2000 Antwerp, Belgium; kdewolf@itg.be (K.D.W.); ideblauwe@itg.be (I.D.); wvanbortel@itg.be (W.V.B.); 5Outbreak Research Team, Institute of Tropical Medicine, Nationalestraat 155, 2000 Antwerp, Belgium; 6Medical Entomology and Zoonoses Ecology Group, Public Health England (PHE), Porton Down, Salisbury SP4 0JG, UK; Alexander.Vaux@phe.gov.uk (A.G.C.V.); jolyon.medlock@phe.gov.uk (J.M.M.)

**Keywords:** exotic mosquitoes, disease vector, public health, vector surveillance, monitoring, globalization, DNA barcoding, real-time PCR, species identification, temperate areas

## Abstract

In Europe, the air-borne accidental introduction of exotic mosquito species (EMS) has been demonstrated using mosquito surveillance schemes at Schiphol International Airport (Amsterdam, The Netherlands). Based upon these findings and given the increasing volume of air transport movements per year, the establishment of EMS after introduction via aircraft is being considered a potential risk. Here we present the airport surveillance results performed by the Centre for Monitoring of Vectors of the Netherlands, by the Monitoring of Exotic Mosquitoes (MEMO) project in Belgium, and by the Public Health England project on invasive mosquito surveillance. The findings of our study demonstrate the aircraft mediated transport of EMS into Europe from a wide range of possible areas in the world. Results show accidental introductions of *Aedes aegypti* and *Ae. albopictus*, as well as exotic *Anopheles* and *Mansonia* specimens. The findings of *Ae. albopictus* at Schiphol airport are the first evidence of accidental introduction of the species using this pathway in Europe. Furthermore, our results stress the importance of the use of molecular tools to validate the morphology-based species identifications. We recommend monitoring of EMS at airports with special attention to locations with a high movement of cargo and passengers.

## 1. Introduction

The introduction of exotic mosquito species into new territories is of major concern to public and veterinary health because of their potential role in the transmission of several pathogens. The geographic spread of these disease vectors is facilitated by international trade and tourism, as well as by climatic and ecological changes. In Europe, during recent decades exotic mosquito species (EMS) such as *Aedes albopictus* (Skuse, 1895), *Aedes japonicus* (Theobald, 1901), and *Aedes koreicus* (Edwards, 1917) have been introduced and have colonised large areas, spreading across multiple countries [1] and, to our concern, the process is still ongoing. 

The main routes for introduction of EMS are the import of commodities such as used tyres or Lucky bamboo, the passive transport of mosquitoes in vehicles (traffic by road, air, and sea), and their natural dispersal from regions where they are established in Europe [2]. EMS surveillance on these introduction routes is therefore critical to promptly detect and control introductions in non-colonized areas.

Nowadays, global aircraft transportation is found to be closely linked to the intercontinental dispersal of vectors and vector-borne diseases [3,4]. Luggage carried in aircraft has occasionally been suspected of harbouring infected *Anopheles* mosquitoes from an endemic area into Europe [5]. In the Netherlands, the air-borne accidental introduction of EMS has been demonstrated by mosquito surveillance schemes at Schiphol International Airport [6,7,8] and demonstrated that the surveillance at airports can be crucial to detect EMS introductions. 

A crucial step in EMS surveillance is the correct species identification of the specimens captured by different sampling methods. Nowadays, the morphological identification of species can be validated by means of DNA-based identification techniques, such as DNA barcoding using the partial mitochondrial cytochrome c oxidase subunit I gene (COI) [9]. For culicids, the technique has repeatedly been used for biodiversity surveys and disease vector identifications in different parts of the world (e.g., [10,11,12,13]). In addition to COI, other DNA fragments have been applied to distinguish among culicid species. The mitochondrial NADH dehydrogenase subunit 4 (Nad4) gene, as well as the nuclear ribosomal internal transcribed spacer 2 (ITS2) are especially useful for the identification of species from the genera *Anopheles* (sole vectors of human malarial parasites) and *Aedes* (vectors of arboviruses and microfilariae) (e.g., [14,15,16]). However, the analysis of DNA fragments generated through (standard) Sanger sequencing can be time consuming and results can usually only be expected after a few days. In order to obtain a rapid identification of (specific) target species (e.g., *Ae. albopictus*, *Ae. japonicus*), faster molecular tools such as real-time PCR assays can be implemented [17,18,19].

In this article we present the results of EMS surveillance at international airports performed by the Centre for Monitoring of Vectors (CMV) of the Netherlands, by the Monitoring of Exotic MOsquitoes (MEMO) project in Belgium, and as part of the Public Health England coordinated invasive mosquito surveillance project. In the Netherlands, EMS surveillance is implemented at Schiphol airport since 2010, and at Rotterdam and Eindhoven airports since 2016. In Belgium, exotic mosquito surveillance at Liège and Zaventem airports has been implemented since 2012 [20,21], except for 2016, whereas at Charleroi airport it started since 2017. In the UK, mosquito surveillance is carried out at the airports of Belfast (Belfast City and Belfast International), Bristol, Cardiff, Doncaster, London Gatwick, Glasgow Prestwick, London Heathrow, Liverpool, Manchester and London Stansted [22]. Mosquito surveillance at London Heathrow and London Gatwick airports commenced in 2010 [22,23,24], and at the other UK airports started in 2016. The morphological identifications of the exotic species have been validated using molecular tools supporting rapid and reliable vector identification, which is the basis for vector surveillance. Early detection of introduced EMS is important for implementing effective measures for mosquito-borne disease control and prevention.

## 2. Material and Methods

### 2.1. Sampling Locations

The airports included in this study are: Amsterdam Airport Schiphol (IATA: AMS), Liège Airport (IATA: LGG), and London Heathrow (IATA: LHR) and London Gatwick (IATA: LGW) airports. Amsterdam Airport Schiphol is the main international airport of the Netherlands and it is located about nine kilometres southwest of Amsterdam. According to the Airport Council International (ACI) report of 2019 [25] it is the third busiest airport in Europe in terms of passenger volume, and ranked the 11th in the world. The international Liège Airport is located about nine kilometres west of the city of Liège in Belgium. This airport mainly focusses on cargo transport and it was in 2018 the 8th busiest airport in Europe in terms of air freight [26]. London Heathrow is the busiest airport in Europe by passenger number and it is located about 23 kilometers west of London. London Gatwick, the 10th busiest, is located about 47 kilometers south of London. 

### 2.2. Sampling Methodology

Exotic mosquito surveillance at the Dutch, Belgian and British airports mainly consists of placing mosquito traps, and performing manual larval sampling (only in Dutch and Belgian airports). Nonetheless, sampling methodologies differ between countries. The number of traps controlled each year at Schiphol, Liège (both latter airports with reported detections of EMS) and London airports (Heathrow and Gatwick) are presented in Table 1. 

At Schiphol airport, EMS surveillance is implemented during the whole year. From November until April, sampling is only performed indoors, while from May to October, traps are placed both indoors and outdoors. Monitoring mainly consists of placing mosquito traps at the most likely locations for inadvertent introduction of invasive *Aedes* mosquitoes at the airport such as: platforms of arrival gates with high volumes of intercontinental flight landing, platforms of gates for European flights arriving from areas where invasive mosquitoes are established (e.g., Italy, southern France), indoor locations where the containers with suitcases are unloaded, and temporary storage locations for arriving cargo, imported fruits, vegetables, or animals. Inside, the traps are placed at locations where sealed containers carrying baggage and cargo are opened for the first time after arrival. Outside, the traps are placed in the vicinity of the locations where aircraft are opened (next to the gates). Traps were continuously operated (24/7) and placed in shaded, wind-protected moist areas. Specifically, the monitoring methods used are:

BG−Mosquitaire traps (BG−M) (Biogents AG, Regensburg, Germany) [27] controlled fortnightly. These traps have been specifically developed for capturing *Aedes* mosquitoes (*Ae. albopictus*, *Ae. aegypti* and related species) and use a patented mix of artificial skin emanations (BG−Sweetscent), in combination with air convection and light-and-dark contrasts.

Standard oviposition traps (OT) controlled fortnightly. The trap consists of a black plastic container (12.5 cm high and 14 cm diameter) filled with water. A floating piece of polystyrene serves as oviposition substrate. If placed outdoors, the top of the OT is covered with stainless steel bird netting to prevent the polystyrene from being blown away by the airplane engines or wind.

BG Gravid *Aedes* Trap (BG−GAT) (Biogents AG) [28] controlled fortnightly. This trap uses water and organic material to create attractant cues for ovipositing female mosquitoes. After being attracted into the trap, mosquitoes cannot reach the water in the trap and typically get stuck on an adhesive panel inserted into the trap. 

Larval sampling using fine mesh aquarium nets (e.g., drainage holes), and adult sampling using mouth aspirators (pooters) were performed if EMS were detected in any of the deployed traps. Larval sampling nets are emptied in a white tray/bowl, and the larvae are collected in tubes using a pipette. 70% Ethanol is added to the tubes to kill and preserve the larvae. These two strategies did not follow a regular scheme but were taken on an ad hoc basis.

At Schiphol airport, if an EMS is identified, an intensive surveillance action follows for at least four weeks. The frequency of the inspections is increased to weekly collections. Furthermore, additional larval and adult searches are implemented in the vicinity of the collection site, and if necessary, additional traps are deployed. The main objective of this intensive surveillance is to evaluate if the detection is an accidental introduction, or if an EMS is reproducing or established at the airport or in the surrounding area. After four weeks without EMS detections, the frequency of the inspections returns to fortnightly and additional deployed traps are removed. 

Collected specimens (eggs on polystyrene, larvae and adults) are sent in isolated sealed plastic bags to the laboratory of the National Reference Centre from the Food and Consumer Product Safety Authority (NVWA/NRC) for morphological and molecular identification. All data from each sampling location is added into VecBase [29] (database of the Centre for Monitoring of Vectors - CMV).

At the Liège airport, the surveillance season starts in April or May, and ends in October or November, each year. Several monitoring methods were used on site:

Mosquito Magnet^®^ Independence (MM trap) (Woodstream^TM^ Co., Lititz, PA, USA), controlled fortnightly. The MM trap produces a continuous stream of CO_2_, heat and moisture into the air, while at the same time the counter flow system sucks the biting insects into a net where they die from dehydration. Octenol is added as a lure, and this trap has been evaluated successfully against a variety of mosquito genera and species for trapping and surveillance of culicids [30,31]. It has been successfully used during past nationwide monitoring of Culicidae in Belgium and the Netherlands [32,33]. Selection of the location for placing the traps at the Liège airport is based on the expected risk for unloading EMS with the arriving cargo. Following this strategy, the trap was placed inside a hangar next to the transport band where cargo is unloaded. 

OT were controlled every four weeks. Hay infusion was used as a lure in 2017. These traps were placed both indoors, close to the transport band, and outdoors, along the platform where aeroplanes land and are opened.

Larval samplings performed every eight weeks at potential breeding sites (e.g., drainage holes, plastic and metal vessels, plastic sheets, tyres) in which mosquito larvae can develop. 

Specimens are transported from Liège to the Institute of Tropical Medicine (ITM−Unit of Entomology) for morphological identification, and afterwards selected specimens are sent to the Royal Museum for Central Africa (RMCA−BopCo) for molecular validation. Collected adult specimens are stored dry in a −20°C freezer before morphological identification, afterwards they are stored dry at room temperature for long term preservation. Larvae are transported alive to the laboratory in vials, and then killed by a thermal shock with hot water (70°C), and finally transferred in 80% (endemic specimens) or absolute (exotic specimens) ethanol. The polystyrene piece with eggs is transported to the lab in a sealed plastic bag. The eggs are then transferred in absolute ethanol and stored at room temperature. All data from each sampling location is stored into VecMap [34].

At the UK airports surveillance is conducted from April to November. Each airport runs a minimum of five BG−GATs, placing the traps in locations on or adjacent to the airport such as along boundary fences and at cargo warehouses, baggage terminals, aircraft gates and dockside locations. Where possible BG−Mosquitaire^®^ or BG−Sentinel^®^ (Biogents AG, Regensburg, Germany) traps are also used, together with Sweetscent^®^ lures (Biogents AG). Traps are checked every two weeks and samples are sent to Public Health England’s Medical Entomology group for morphological identification.

## 3. Mosquito Species Identification

### 3.1. Morphology−Based 

In the laboratory, mosquitoes are sorted from other trapped insects, counted and morphologically identified using, among others, the keys of Schaffner et al. [35], Becker et al. [36], Gunay et al. [37], Darsie and Ward [38], and Gillies and Coetzee [39]. Each mosquito identified as exotic was double-checked by a second culicid taxonomist of each respective institution. In our study, a sample represents the collection of specimens after a visit. Samples can contain mosquito specimens (present), or be empty or only containing other arthropods (absent).

### 3.2. DNA-Based 

Molecular identification of exotic *Aedes* species found in Dutch airports was performed using a real-time PCR assay, especially developed by the CMV surveillance for the identification of *Ae. albopictus*, *Ae. aegypti*, *Ae. atropalpus*, and *Ae. japonicus.* The used methodology and protocols of the assays were as described in van de Vossenberg et al. [19], and the identification is based on the investigation of the species specific polymorphisms of the nuclear ribosomal internal transcribed spacer 1 (ITS1) and the COI regions. This analysis was restricted to damaged adult specimens missing morphological features, and to *Aedes* eggs. 

For the other identified exotic Culicidae genera (non-*Aedes*) found in the Netherlands and Belgium, standard Sanger sequencing of three different DNA regions (COI, Nad4 and ITS2) was performed to validate the morphological identification results. Amplification of the 658 bp COI DNA Folmer region [40] was performed on selected specimens as described in the EPPO PM7/129(1) publication [41]. Primers and PCR cycling conditions for the amplification of the three DNA regions are displayed as Appendix A [41,42,43]. All amplifications were performed in a 20 µL reaction mixture containing 2 µL of DNA template (regardless of initial concentration), 2 µL of 10× buffer, 1.5 mM MgCl_2_, 0.2 mM dNTP, 0.4 µM of each primer, and 0.03 units/µL of PlatinumTM Taq DNA Polymerase (Invitrogen^TM^, Waltham, MA, USA). PCR products and negative controls were checked on a 1.5% agarose gel, using a UV transilluminator and the MidoriGreenTM Direct (NIPPON Genetics Europe, Dueren, Germany) method. Positive amplifications were subsequently purified using the ExoSAP-ITTM protocol (following manufacturer’s instructions) and sequenced in both directions on an ABI 3230xl capillary DNA sequencer using BigDye Terminator v3.1 chemistry (ThermoFisher Scientific, Waltham, MA, USA).

The quality of the sequencing output was checked with Geneious^®^ R11 (Biomatters Ltd., Auckland, New Zealand): paired bi-directional strands were trimmed, corrected, assembled, translated into amino acids to check for stop codons (if coding region), and consensus sequences were extracted for each specimen and each DNA marker. Subsequently, COI consensus sequences were compared against the Identification System of BOLD, with Species Level Barcode Records option (www.boldsystems.org). Nad4 and ITS2 consensus sequences were compared against the BLAST web application of GenBank (https://blast.ncbi.nlm.nih.gov/Blast.cgi).

To perform further analyses, a comprehensive list of mosquito species occurring in Belgium and in the Netherlands was established [11,32,33,44,45,46,47,48,49,50,51], to which we added the newly identified EMS (information in Appendix A). The analyses were only performed using COI since ITS2 confirmed the COI species identification results, and the database of available COI sequences from online repositories was much more substantial. Nad4 did not provide conclusive results (Appendix A). 

For each of the species potentially occurring in Belgium and/or the Netherlands, all available COI sequences from BOLD were downloaded, cleaned and aligned using the software Geneious^®^ R11 (Biomatters Ltd., Auckland, New Zealand). Duplicates (i.e., identical sequences) were discarded per species to limit the size of the database. In addition, sequences of less than 300 bp were deleted. For some species, however, the list of available sequences was still substantial and an additional selection was performed. The newly generated consensus sequences of the collected EMS, as well as sequences of an outgroup (*Drosophila,* Genbank accession numbers: MG087305, MG081935 and HM102299), were added to the final dataset before aligning all sequences using ClustalW in Geneious^®^ R11. The final alignment was then trimmed to only retain the 658 bp COI Folmer region. 

Based on Kimura 2−parameter (K2P) distances [52], a rooted Neighbour−Joining tree (NJ) was constructed using MEGA7 [53,54], with branch support assessed by 500 bootstrap replicates [55]. Clustering of the generated sequences in relation to the other species in the dataset was examined. Species identification is considered correct if the generated sequence(s) cluster with all and only conspecific sequences, supported by high bootstrap values. Finally, average interspecific K2P distances, as well as the maximum observed K2P distances between conspecific sequences, were calculated among COI sequences with the package Spider [56] using R software (version 3.6.2).

## 4. Results

### 4.1. Exotic Mosquito Capture

In Schiphol, from 2016 until 2018 a total amount of 3602 samples were collected using traps and larval/adult samplings (Table 2). A total of 13,906 mosquito specimens were collected belonging to five mosquito genera: *Aedes*/*Ochlerotatus* (0.58%), *Anopheles* (1.11%), *Culex* (97.41%), *Culiseta* (0.88%), and *Mansonia* (0.01%). Exotic mosquitoes found at the airport of Schiphol were identified as *Ae. albopictus* (*n* = 5), *Ae. aegypti* (*n* = 69), *Anopheles crucians* (Weidemann, 1828) (*n* = 1), *Anopheles subpictus* (Grassi, 1899) (*n* = 1) and *Mansonia* sp. (Walker, 1848) (*n* = 2) (Table 3).

In Liège, from 2017 till 2018 a total amount of 148 samples were collected using traps and larval sampling (Table 2). A total amount of 40 mosquito specimens were collected belonging to four mosquito genera: *Aedes*/*Ochlerotatus* (5.0%), *Anopheles* (7.5%), *Culex* (82.5%), *Culiseta* (5.0%). At the airport of Liège, only one EMS was caught, namely *Anopheles pharoensis* (Theobald, 1901) (*n* = 1). 

Both exotic *Aedes* species were morphologically identified using the keys of Becker et al. [36]. *Aedes aegypti* was captured using BG-M traps (44.92%), BG-GAT traps (7.25%), OT (43.48%), and live catches with the mouth aspirators (4.35%). During the study, *Ae. aegypti* mosquitoes at Schiphol airport were caught at outdoor and indoor locations, and the captures were spread over the year. From November until April, detection of *Ae. aegypti* (*n* = 7) was registered only in indoor traps at cargo and baggage handling areas, as these traps are the only deployed during this period. From May until October, *Ae. aegypti* (*n* = 55) was found in indoor traps and in additional traps placed outdoors during this period (*n* = 4).

Detections of *Ae. aegypti* in 2016 (*n* = 6) have already been reported by the authors [8], and these specimens were collected exclusively using BG-M traps placed indoors and outdoors. In 2017, *Ae. aegypti* was only found in BG-M traps (*n* = 6) placed indoors at baggage handling and inspection areas. In 2018, the species was again collected in BG-M traps (*n* = 19) placed indoors and outdoors (at two different gates), in OT (*n* = 30), BG−GAT traps (*n* = 5) placed indoors at baggage handling areas, and live catches with the mouth aspirators (*n* = 3). At Schiphol airport, all specimens of *Ae. albopictus* were captured using BG-M traps. *Aedes albopictus* (*n* = 2) was collected in 2017 at two indoor locations (cargo and baggage handling areas). In 2018, the species was found indoors in a trap placed in a cargo handling area (*n* = 1), and outdoors in a trap placed next to a gate entry (*n* = 1), and at the entry in a cargo handling area (*n* = 1).

Following the detection of *Ae. aegypti* in 2016 at Schiphol, the intensification of the surveillance at different baggage handling areas led to the detection of other exotic species. Since diurnal mosquitoes are attracted towards natural light, surveillance included the inspection of window sills at the baggage handling areas where *Ae. aegypti* was found indoors. During two inspections, *An. crucians* (*n* = 1) and *An. subpictus* (*n* = 1) specimens were found dead and dry on windows sills. Both species were found at the same baggage handling location. In 2018, two specimens of *Mansonia* sp. were collected dead, morphologically intact, inside aircraft containers from direct flights originating from Paramaribo (Surinam). The search on aircraft containers was performed as an ad hoc activity after multiple detections of *Ae. aegypti* in 2018. Using the key of Central and South American mosquito species included in Becker et al. [36] they were identified as *Ma. titillans*. To corroborate the species identity of the *Mansonia* specimens, South American mosquito experts (Fabio Castelo Branco Fontes Paes Njaime and Fabio Medeiros da Costa, pers. comm.) were consulted for validation, who also assigned the specimens to *Ma. titillans*.

At the airport of Liège, one single specimen of the exotic mosquito species *An. pharoensis* was collected in 2017 using the MM trap located indoors at the cargo handling area. In the United Kingdom, there were no detections of exotic mosquitoes at London Heathrow or London Gatwick airports. At Heathrow and Gatwick airports, from 2016−2018, a total of 320 indigenous mosquitoes were sampled (see Table 3) all found in the BG-GATs and BG-Mosquitaire traps.

### 4.2. Species Identification Validation

From 2016 to 2018, 25 exotic *Aedes* specimens from Schiphol were submitted to real-time PCR analysis to confirm the morphological identification. The morphological and molecular identification were in line: four specimens were identified as *Ae. albopictus* and 21 specimens as *Ae. aegypti*.

The morphological identifications were confirmed for three specimens of the non-*Aedes* species, by comparing the generated COI and ITS2 sequences to sequences extracted from the online repository BOLD and GenBank. The obtained similarity percentages for these three specimens ranged from 99.08 to 100% (Appendix A). For the two *Mansonia* specimens, DNA identifications were inconclusive, with none of the DNA markers matching with high similarity percentages on BOLD or GenBank. The generated sequences for the five specimens were deposited in GenBank with accession numbers: MT329066-MT329070 for COI, MT366208-MT366212 for ITS2 and MT334776-MT334779 for Nad4.

In Belgium and the Netherlands, 45 species from six genera occur (Appendix A). In total, 2718 COI sequences extracted from the online repositories for these 45 species were retained in the final analyses (Min: 1; Max: 437; Mean: 60 sequences per species). Pairwise K2P distances between species varied from 0.30 to 13.76%, and the largest intraspecific K2P distance was 14.72% for *An. subpictus* (Appendix A). The divergence between Culicidae species averaged 6.36%. An overlap between the intra- and interspecific K2P divergences of congeneric sequences was observed (Appendix A), but did not concern any of the EMS. In the NJ-tree based on COI sequences, most species formed well-supported clusters with high bootstrap support (BS; Figure 1). 

The four exotic specimens (excluding *Mansonia* sp. specimens) caught at Belgian and Dutch airports are highlighted in grey on the NJ−tree. The sequences of *An. crucians* and of *An. pharoensis*, form monophyletic clusters with maximum BS support, incorporating the sequences of the query specimens. *An. subpictus* sequences, including the produced sequence, cluster in three distinctive groups, all with maximum support. 

Finally, the morphological identification of the *Mansonia* specimens could not be confirmed using DNA−based techniques. All COI sequences from genus *Mansonia* were extracted from BOLD (227 COI sequences, representing 11 species), and the clustering support was investigated by a NJ−tree analysis (Appendix A).

## 5. Discussion

In this article we report on the aircraft facilitated transport of exotic mosquito species (EMS) to two international airports, one each in Belgium and the Netherlands. In the UK, during our study no exotic mosquitoes were found although *Ae. albopictus* has been detected at sites supporting vehicular transport [57]. For Belgium, this is the first time that EMS are reported from airports. However, the passive transport of mosquitoes in aircraft has been demonstrated in the past in the Netherlands. In 2008, mosquitoes collected by the flight attendants on a flight from Dar es Salaam (Tanzania) to Schiphol were molecularly identified as *Culex quinquefasciatus* Say [6]. In 2010 and 2011, the EMS *Cx. quinquefasciatus*, *Culex antennatus* and *Aedes mcintoshi* were intercepted on 10 of 38 inspected aircraft [7], while in 2016, six *Ae. aegypti* were captured at baggage areas at the airport [8]. Based upon these findings and given the high volume of air transport movements per year, the establishment of EMS after inadvertent introduction via airplanes has been considered a potential risk.

We hypothesize that the exotic mosquitoes captured in this study were brought in through the containers used for luggage transport or cargo because they were all captured in traps near baggage handling areas. We assume that the mosquitoes were not affected by the WHO recommendations for disinsection for aircraft (assuming that these recommendations were implemented) [58] because in most of the cases they were alive upon arrival and could reach the traps. The intercepted mosquitoes could have arrived from different geographic regions around the globe. The intercepted *Aedes* species (*Ae. albopictus* and *Ae. aegypti*) have populations in all continents except Antarctica due to their invasive potential. The other detected species have more limited distributions: the suspected *Mansonia spp.* (*Ma. titillans*) is restricted to North and South America [59], *Anopheles crucians* complex is distributed in North and Central America [60], *An. subpictus* to Asia and Oceania [61], and *An. pharoensis* to Africa and surroundings countries [62].

The EMS surveillance methodology that was applied at the airports seems to be adequate to detect the accidental introduction of EMS. At Schiphol, the number of traps deployed varied over the years and within the years (Table 1) and traps are placed at strategic locations for capturing arriving EMS. In Liège, trap locations remained constant over years. In 2017, the Mosquito Magnet (MM) trap detected the first accidental introduction of an EMS in Liège, *An. pharoensis*. At Schiphol airport, BG-M traps placed indoors and outdoors, were effective to detect both *Ae. albopictus* and *Ae. aegypti*, and BG-GAT and OT placed indoors detected *Ae. aegypti*. Lure used in the BG-M traps has proven to be efficient for capturing adult *Ae. albopictus* and *Ae. aegypti* (both males and females) in the field [63,64,65]. Since these traps are especially designed for trapping *Aedes* species, the BG-M traps could be considered suboptimal for detecting other EMS species such as the exotic *Anopheles* species found in Schiphol on the window sills or inside the aircraft containers. However, from 2016 until 2018 more than 80 indigenous *Anopheles* specimens were captured at Schiphol in BG-M traps demonstrating also some capabilities of this trap for capturing *Anopheles* species.

Human trade and transport have contributed to the global spread of *Ae. aegypti,* a species known to transmit, among other arboviruses, dengue, yellow fever, chikungunya and Zika viruses [66,67,68,69]. Introductions of *Ae. aegypti* in the Netherlands have been detected by the authors at used tyre companies and at the Schiphol airport [8,70,71]. In comparison to 2016 and 2017, there is a remarkable increase of captured *Ae. aegypti* specimens at the airport in 2018. Over the period 2016−2018 the same number of locations were surveyed and after each detection of an EMS extra traps were placed at the locations for four weeks to detect new introductions or possible establishment of the species. In 2018 the number of captures was higher, probably indicating a higher number of flights importing the species. Also, in 2018 we found *Ae. aegypti* using OT and BG-GAT traps. These traps (OT and BG-GAT) are deployed in the premises of the airport to detect possible reproduction of introduced EMS. In 2018, larval sampling in other human-made habitats were negative, suggesting OT and BG-GAT traps might be more attractive or the first to be encountered by the gravid female mosquitoes upon arrival. Eggs and hatched larvae (but no pupae) could be collected in these traps in 2018, which suggests that in a two week period, *Ae. aegypti* females could lay eggs, the latter hatching and developing into larvae. This fact stresses the importance of having an adequate sampling frequency at surveillance sites when using OT. Depending on the temperatures registered at sampling sites, to avoid proliferation, the sampling frequency when using OT should increase to weekly collections as for example in Mediterranean regions. Based on our field observations, in the countries included in our study, we recommend biweekly collection scheme when using OT, or the use of larvicides to avoid the proliferation of EMS.

Like *Ae. aegypti*, *Ae. albopictus* has expanded its distribution range due to global trade and transport [72]. *Aedes albopictus* is an EMS that causes nuisance by its biting behaviour [73] and has been proven to transmit more than 22 viruses under laboratory conditions [74]. In the field, it is a competent vector of chikungunya and dengue viruses [75], and probably Zika virus given the local transmission of Zika in France [76]. To our knowledge, the detections of *Ae. albopictus* at Schiphol are the first evidence of accidental introduction of the species using this pathway in Europe. Due to its wide distribution range, and its association with human activities, *Ae. albopictus* could probably be present in the vicinity of airports. Interestingly, the present study highlighted that the species was captured on a variety of locations at the airport related to cargo handling, sorting luggage, or aircraft arrivals. The repeated introductions of the species at Schiphol airport is of great concern. The species has demonstrated its hitchhiking behaviour and invasiveness potential in Europe [77], and, on the contrary to *Ae. aegypti*, the climate of the Netherlands is considered relatively favourable for the establishment of *Ae. albopictus* [78]. However, as the specimens were captured in traps and not directly on the aircraft, the origin of the captured specimens remains unknown. 

*Mansonia spp.* has a nearly worldwide distribution, but the *Mansonia* sp. were collected dead and intact inside aircraft containers from direct flights originating from Paramaribo (Surinam). In the Americas, mosquitoes of the *Mansonia* genus are related to forested areas, but they can be frequently found in urban and peri-urban areas [79] where they can be aggressive biters. In northern Surinam, *Ma. titillans* is present in typical coastal areas [80], and in Paramaribo, it is reported that large numbers of *Ma. titillans* enter houses [81]. Using the key of Central and South American mosquito species included in Becker et al. [36] it was identified as *Ma. titillans*. However, it cannot be fully excluded that it might concern another *Mansonia* sp. such as *Ma. pseudotitillans, Ma. nigricans,* or *Ma. venezuelensis*, which are present in the area of origin of the aircraft and are morphologically similar [82]. To corroborate the species identity of the specimen, mosquito experts in South American mosquitoes were consulted for validation since DNA markers did not allow a molecular identification, certainly because of the limitations of the online DNA reference databases. As shown in the results, the *Mansonia* specimens found were not molecularly similar to those in the DNA reference databases, suggesting that thorough taxonomic clarification is necessary for this taxon in order to explain the differences found. Furthermore, the two produced barcodes from the caught specimens are also extremely different (with more than 60 polymorphic sites between the COI sequences), which would be indicative of even two different *Mansonia* species. DNA-barcoding is a powerful tool and is especially useful when diagnostic morphological features are damaged during the process of collection or storage, or when the species has very little or no morphological characteristics differentiating it from related species. However, for DNA-barcoding to produce accurate species identifications, a reliable and comprehensive barcode reference library is needed, with which the query sequences can be compared. In the case of genus *Mansonia,* the number of represented species is limited, as well as the number of available DNA sequences per species, and the sampling coverage of the barcoded specimens in regard of the overall distribution range of each species. These gaps can lead to an inconclusive molecular identification.

The *An. crucians* mosquito found dead on a window sill in Schiphol was morphologically identified as a member of the *An. crucians* group, and molecularly identified as *Anopheles crucians*. *Anopheles crucians* Wiedemann is a common species in the southeastern and midwestern United States, and can also be found in Central America and the Caribbean islands [83]. *Anopheles crucians* is a species complex that consists of *An. bradleyi*, *An. georgianus* and *An. crucians s.s.* [83]. The adults of the three species are difficult to separate, and only the larval and pupal stages can be morphologically identified to species level [84]. Furthermore, *An. crucians s.s.* is also a complex that can be separated using rDNA ITS2 sequences, and including the species A, B, C, D and E [83]. The adult specimen collected at Schiphol was molecularly identified as species E. *Anopheles crucians* develop in permanent or semi-permanent freshwater pools, ponds, streams, swamps or along lake margins. The water may be acid or alkaline, although acid water seems to be preferred [84]. The females frequently bite outdoors at night or even during the day in the woods [85]. Females take their blood meal from a variety of mammals such as humans, cattle, rabbits and deer [86]. Adults of *An. crucians* don’t often go into houses, they rather look for cellars or barns to rest [87]. Several viruses were found in *An. crucians* including Eastern Equine Encephalitis (EEE), LaCrosse and West Nile viruses (WNV) [85,86,88]. The filarial nematode *Dirofilaria immitis* has also been detected by polymerase chain reaction in *An. crucians* from Georgia, USA [89]. *Anopheles crucians* can be a good vector of *Plasmodium falciparum* and *P. vivax*, both of which causing malaria in humans [90,91]. Infection of *An. crucians* with *P. falciparum* and *P. vivax* has been reported in Florida and Louisiana in the past [84] but it is not entirely clear whether *An. crucians* makes an active contribution to the spread and outbreaks of malaria [85,86].

*Anopheles* (Cellia) *subpictus* Grassi also is a species complex. The subpictus complex has a wide distribution, ranging from northeastern Pakistan, across India, Sri Lanka, Bangladesh, Myanmar, Thailand and along coastal regions of southern Cambodia, Vietnam and coastal areas of Malaysia, Indonesia, Timor-Leste, Papua New Guinea and extending as far East as the Solomon Islands [92]. The subpictus complex is currently considered to include four sibling species, named species A, B, C and D. Species A, C, and D are generally found in fresh-water, and species B restricted to coastal brackish-water habitats [92]. Natural larval habitats for members of the complex include lagoons, shallow ponds, marshes, slow-flowing rivers, natural pools and margins of small streams. Larvae have also been collected from small, artificial containers. Members of the subpictus complex are generally zoophilic (including humans), with no clear preference for either indoor or outdoor biting. The subpictus complex is considered a primary vector of malaria in many Southeast Asian countries and is regarded as a secondary vector in Sri Lanka [93], but further work is needed to confirm the vectorial capacity and distribution of each species across the wide geographical range of the complex.

*Anopheles pharoensis* is a tropical mosquito species mostly present in West, East and southern Africa, Egypt, Israel, and Syria [62,94]. It prefers marshes, swamps, rice fields and ponds, especially those with abundant grassy or floating vegetation [95]. The larvae are primarily found in large vegetated fresh water swamps, but also breed along lake shores and among floating plants such as *Pistia* and *Potamogeton*. They are also found in reservoirs, rice fields, streams, ditches and overgrown wells [96]. The females feed predominantly on domestic animals, especially bovids [39,97], but enter houses readily at night and bite humans. The main daytime resting sites are outdoors, particularly in the vegetation and they start biting in large numbers soon after dusk, the attack continuing at a high level until midnight [96]. It can disperse long-distances from nine to 70 km [95,96,98]. The species is adapted to dry environmental conditions [99]. *Anopheles pharoensis* is a well-known vector of malaria in Egypt [36]. In African countries it is a common secondary vector of malaria [95,100]. The finding of this species at the airport of Liège confirms that EMS can enter via cargo transport [58], however the ecological preference of this EMS are dry environments, consequently the climatic conditions in Belgium are considered as non-suitable for potential/possible establishment.

The detection of exotic *Anopheles* species at Belgium and Dutch airports demonstrates their ability to move from areas of origin using air traffic as pathway. However, detection of exotic *Anopheles* is not new at airports, and even malaria cases caused by infected non-indigenous *Anopheles* at European airports, have been registered [4,5,101]. The authors consider that it is unlikely that exotic *Anopheles* species will establish in Belgium or the Netherlands, due to their ecological and climatic requirements.

In this study, EMS were only found at the airports of Liège and Schiphol. At other Dutch (e.g., Eindhoven, Rotterdam), Belgian (e.g., Charleroi, Zaventem) and British airports (e.g., London Heathrow, London Gatwick) mosquito surveillance was also implemented but no EMS were detected. The reasons for the multiple EMS captures at Schiphol airport may be attributed to the large total volume of passengers and cargo transported, and to the larger relative sampling effort applied during the study, compared to the sampling effort at London and Liège airports. Schiphol airport is by far the busiest airport in the Netherlands, transporting almost 90% of the total volume of passengers [102]. In 2018, Schiphol had the third-highest number of passengers, the third-highest volume of cargo and ranked second in number of air transport movements in Europe [103]. On the other hand, London Heathrow is the busiest airport in Europe and still no EMS have been detected during surveillance. In Belgium, the airport of Liège is the largest cargo airport and ranked 8^th^ in Europe in 2018. Goods are imported from Ethiopia, Uganda, Kenya, USA, Qatar and Israël. In 2017 the airport of Liège had a record year with 717,000 tonnes of goods handled [104] and this growth continued in 2018 [26], which will increase the risk of importing EMS through this airport.

As shown by our results, and the results of many other studies, the transport of mosquitoes in aircraft is not unusual, and concern about the possible transportation of mosquitoes carrying pathogens has led to recommendations for aircraft disinsection [58]. Disinsection [105] is the procedure whereby health measures are taken to control or kill the insect vectors of human pathogens present in baggage, cargo, containers, conveyances, goods and postal parcels. In the review of Mier-y-Teran-Romero et al. [106], researchers concluded that each aircraft could transport one mosquito specimen. However, the study claims that the probability of introduction of a pathogen by mosquitoes is low in comparison with the probability of introduction via infected human travelers. This is because mosquitoes are rarely found on aircraft, they are unlikely to be infected, and the mosquito must survive long enough to complete the incubation period and feed on another human [106]. The use of an effective disinsection programme may reduce the threat of EMS accidental introduction through air transport into areas where those species are not already established [107]. The authors are not aware of the disinsection procedures applied on the flights arriving at Schiphol and Liège, or if the disinsection also included the luggage containers or cargo. However, looking at our findings, EMS specimens appear to effectively enter aircraft, possibly avoiding disinsection procedures, and are transported alive to Europe. As far as the authors know, on the contrary to aircraft transporting passengers, no insecticide or treatments are implemented on cargo airplanes coming from outside Europe. For Europe, phytosanitary regulations do not include obligatory insecticide treatment of cargo space when flowers, fruit or vegetables are transported to Europe. Considering the results of this study, imported cargo is believed important for introduction of EMS, and disinsection and surveillance of this Point of Entry is highly recommended.

## 6. Conclusions

The findings of our study demonstrate the aircraft mediated transport of EMS into Europe from a wide possible geographical range. From the origin areas, EMS travelled into the aircraft and left the aircraft or the transported containers after landing upon opening of the doors. Our findings also demonstrate the value of EMS surveillance programs in order to detect and engage actions against possible establishment of EMS and airport disease transmission. 

The morphological identifications of *Ae. aegypti*, *Ae. albopictus*, *An. crucians*, *An. subpictus* and *An. pharoensis* were confirmed by molecular methods. Yet, the morphology-based identification of the *Mansonia* specimens could not be validated using DNA sequences. The results obtained stress the importance of the use of molecular tools to validate the morphology-based species identifications, and vice versa, and the necessity of filling the gaps of DNA reference databases for the identification of species (e.g., *Mansonia*).

Even if the risk of vector-borne disease transmission caused by imported mosquitoes is considered low, monitoring of EMS must be enforced at airports to avoid possible establishment of new EMS in Europe with special attention to locations with a high movement of cargo, passengers and military equipment.

## Figures and Tables

**Figure 1 ijerph-17-03450-f001:**
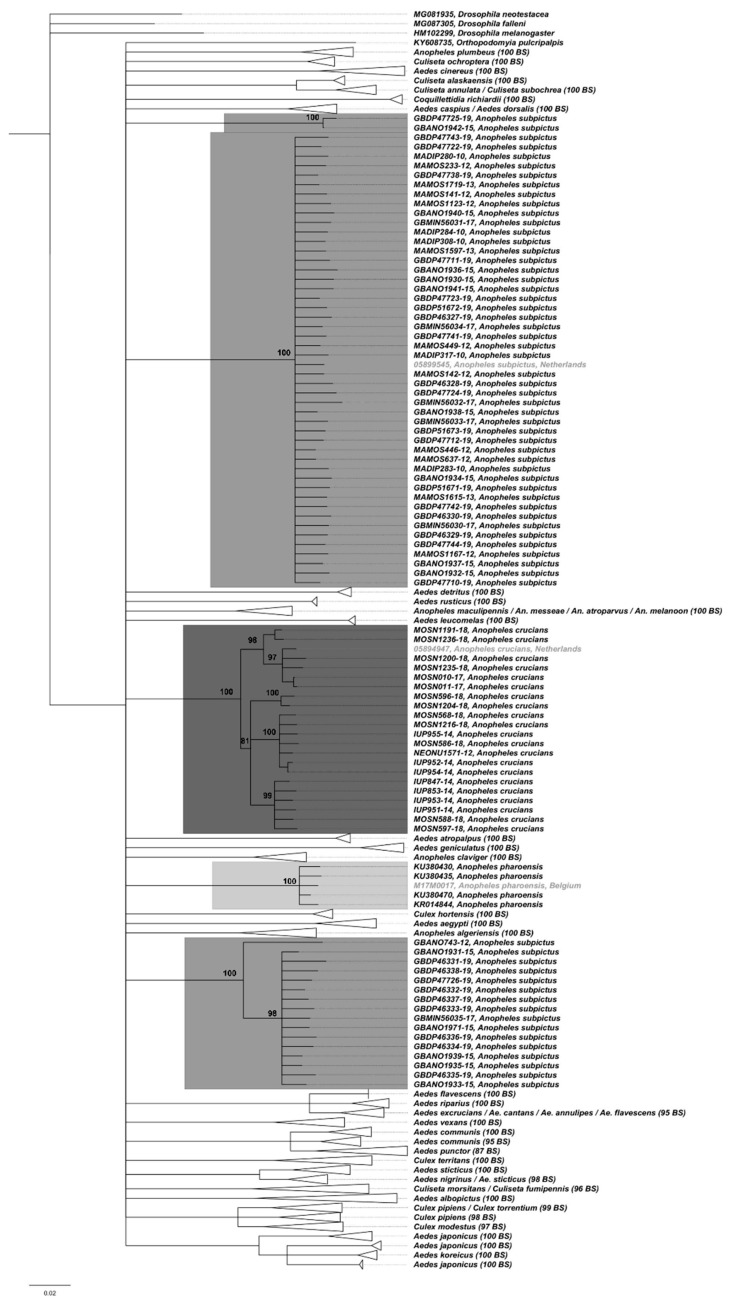
Neighbour-Joining tree including 45 Culicidae species recorded in Belgium and in the Netherlands based on COI (K2P; 658 bp fragment; 2721 barcodes (including the three newly generated sequences)). The bootstrap values (BS; 500 replicates) are shown in the tip labels for the collapsed sequence clusters, while they are displayed at the branch points for the non-collapsed species (sequences of query specimens highlighted in grey). Clusters were collapsed to make the overall tree visually clearer. Minimum bootstrap displayed is 75, other branches are collapsed.

**Table 1 ijerph-17-03450-t001:** Number of mosquito traps placed each year at the airports of Schiphol (the Netherlands), Liège (Belgium), and London (Heathrow and Gatwick, United Kingdom). MM Trap = Mosquito Magnet trap; BG-M = BG Mosquitaire; OT = Oviposition trap.

Trap Type	2016			2017			2018		
	Schiphol	Liège	London	Schiphol	Liège	London	Schipho	Liège	London
**MM trap**	-	-	-	-	1	-	-	1	-
**BG-GAT**	-	-	20	-	-	20	20	-	20
**BG-M**	31	-	2	35	-	2	48	-	2
**OT**	31	-	-	35	10	-	22	10	-

**Table 2 ijerph-17-03450-t002:** Number of samples collected by sampling method at Schiphol (the Netherlands), Liège (Belgium) and London (Heathrow and Gatwick, United Kingdom) airports. n.a.: not applicable. Numbers in brackets represents the mean and standard error of samples containing EMS of each trap type. MM Trap = Mosquito Magnet trap; BG-M = BG Mosquitaire; OT = Oviposition trap.

Sampling Method	2016			2017			2018		
	Schiphol	Liège	London	Schiphol	Liège	London	Schiphol	Liège	London
**MM trap**	n.a.	n.a.	n.a.	n.a.	7 (0.142 ± 0.142)	n.a.	n.a.	14	n.a.
**BG-GAT**	n.a.	n.a.	260	n.a.	n.a.	260	477 (0.013 ± 0.006)	n.a.	260
**BG-M**	446 (0.013 ± 0.005)	n.a.	52	874 (0.009 ± 0.003)	n.a.	52	1261 (0.015 ± 0.003)	n.a.	52
**OT**	358	n.a.	n.a.	423	40	n.a.	151 (0.007 ± 0.005)	70	n.a.
**Larval and adult sampling**	49	n.a.	n.a.	11	4	n.a.	29	13	n.a.
**Total samples**	853	0	312	1308	51	312	1441	97	312

**Table 3 ijerph-17-03450-t003:** Number of specimens collected at Schiphol (the Netherlands), Liège (Belgium), and London (Heathrow and Gatwick, United Kingdom) airports. (*) Published in [8]; A: adult, E: egg, L: larva, P: pupa. Results on non-indigenous species are indicated in bold.

Location	Species	Year			Total
		2016	2017	2018	
**Schiphol**	*Aedes aegypti*	6A (*)	6A	28A; 16E; 13L	40A; 16E; 13L
	***Aedes*** ***albopictus***		**2A**	**3A**	**5A**
	*Aedes/Ochlerotatus indigenous*	1A; 1E		5A	7A
	***Anopheles crucians***	**1A**			1A
	***Anopheles subpictus***	**1A**			1A
	*Anopheles indigenous*	25A	35A	93A	155A
	*Culex indigenous*	2049A; 495E; 844L; 40P	2564A; 20E; 3L	7157A; 209E; 19L	11,770A; 724E; 866L; 40P
	*Culiseta indigenous*	21A	36A	65A	122A
	***Mansonia sp.***			**2A**	**2A**
**Liège**	***Anopheles pharoensis***		**1A**		**1A**
	*Anopheles indigenous*		2A		2A
	*Culex indigenous*		11A	19A; 3L	30A; 3L
	*Aedes indigenous*			2A	2A
	*Culiseta indigenous*		1A	1A	2A
**London**	*Culex indigenous*	26A	2A	288A	316A
	*Anopheles indigenous*			1A	1A
	*Culiseta indigenous*	2A		1A	3A

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
