# Peer review of "Detection of Exotic Mosquito Species (Diptera: Culicidae) at International Airports in Europe"

_ijerph, 2020, doi:10.3390/ijerph17103450_

Round 1

Reviewer 1 Report

General Comments

The paper addresses relevant questions on the possibility of exotic mosquito species (EMS) introduction via aircraft in European airports.

The object of the study is of great interest and the data presented is significant and informative, spanning several years of collection and including both taxonomic and molecular validation of insect identifications.

There are some issues I’d like to point out in hope they help the authors to improve the quality of the work:

The Material and Methods Section should be revised to clearly define the specific sample locations and methodology used for the data that is actually reported in the results section.

Regarding the presentation of results, I believe it could be greatly improved by addition of at least a graphical resource (e.g. barplot). Only two tables are shown and some additional data is directly presented in the text. The total number of captures is shown in Table 2 and the distribution of species per year and location is presented in Table 3. However, only partial results on trap type performance for every insect species is presented in results and discussion. A deeper analysis and representation of the existing data would, in my opinion, improve the readability and reinforce the relevance of the study. Discussion of differential sampling effort and airport size on detection sensitivity is specially reccomended.

As for the discussion, some results are presented for the first time in this section, which should be introduced earlier. I believe that a more systematic analysis of the results might provide more sound or even additional conclusions and suggestions.

The discussions need to be fully supported and properly analyzed in relation to the presented data. For instance: “The main reason for the multiple EMS captures in The Netherlands at Schiphol can be attributed to the large total volume of passengers and cargo transported.” Although such a statement might be true, the discussion should also include the aspect of comparative sampling effort (maybe adjusted for airport size).

Specific comments:

Lines 77-94. This section initially leads the reader to believe that all mentioned airports are included in this study, while data for only three airports is provided in the results section. Such contextualization on current European airport sampling should be stated in the introduction, not in M&M.

Line 117. “Standard oviposition traps (OT)” instead of “Standard oviposition (OT) traps”.

Line 144. I believe the registered trademark is Mosquito Magnet, not the company: “Mosquito Magnet® Independence (MM trap) (WoodstreamTM Co., Lititz, USA)”. Please check.

Line 153: Suggestion: “OT were controlled every four weeks. Hay infusion was used as a lure in 2017”

Line 153: Is there an expected decrease in attractiveness due to lack of attractant in years other than 2017?

Line 188: I’d suggest a new paragraph for this sentence.

Line 189: Suggestion: “sequencing of three different DNA regions (COI, Nad4 and ITS2)”

Lines 190-199: Cycling conditions are made explicit for only one DNA region. Please consider adding a table with PCR region, primers, cycling conditions and references. Also, try to clearly discern when talking about Amplification and Sequencing.

Line 262: Both Ae. aegipti an Ae. albopictus captured only in BG-M traps. Reporting the detail of captures depending on trap type would help. It might also lead to discussion on trap type pros and cons and possible specificities.

Line 279: Do you think the trap type is relevant?

Line 288: I’d suggest to use colon (were in line: four specimens...)

Line 295: Missing Accession Codes.

Lines 322-324: Besides published data, the findings of the present paper in UK airport should be more clearly stated.

Line 333-335: No temporal data is provided. When where they captured? During winter only indoor traps (all of them in cargo areas) were deployed, so this information could be relevant, as traps out of baggage area seem to be deployed only during summer.

Lines 347-350: This information could be detailed in M&M, as it does not seem to add to the discussion.

Line 355: please avoid repetition: “Since these traps are especially designed for trapping Aedes species, they could…”

Line 357-359: The fact that a trap is able to capture some insects seems to be presented here as demonstrative of its efficiency. However, insect capture by itself is not proof that the trap is not suboptimal (i.e. less efficient than a more specific trap type). Since evidence is not provided in that sense, please reword the sentence.

Line 363-364: I believe that these results have not been mentioned in the Results section.

Line 366: In Table 1, 2018 shows a sample number increase in Shiphol of about 70% (1441 vs 853). I would not say the sampling effort has not changed.

Line 367: The placement of extra traps is not mentioned in M&M. Please do so.

Line 369: It is not clear how the trap type is relevant for this conclusion. Please clarify or remove the mention to trap type.

Line 370: Please clarify if the reference of ‘These traps’ is meant for both trap types (OT and BG-GAT) or only OT.

Lines 376: What is considered a 'good sampling frequency'? Sampling frequency is not mentioned in M&M. Justify the reasons that support the recommendation for biweekly sampling.

Line 383: This information on Aedes first report might be better in the introduction.

Line 428: it does not make much sense to refer to the development stage of a species in a molecular identification result. Please reword (e.g: The immature specimen collected…)

Lines 477-478: I believe the sampling effort should also be introduced into discussion here. Schiphol did 14 times more samples than London! A relative effort should be considered.

Table 1. Please check formatting of column labels.

Table 3. Bold type seems to indicate non-indigenous species but, if so, it should be clearly explained in the table caption. Also, Anopheles crucians is not in bold for 2016 data point (is it a mistake?). Please check format: the term ‘indigenous’ is not consistently italicized.

Author Response

General Comments

The paper addresses relevant questions on the possibility of exotic mosquito species (EMS) introduction via aircraft in European airports.

The object of the study is of great interest and the data presented is significant and informative, spanning several years of collection and including both taxonomic and molecular validation of insect identifications.

There are some issues I’d like to point out in hope they help the authors to improve the quality of the work:

The Material and Methods Section should be revised to clearly define the specific sample locations and methodology used for the data that is actually reported in the results section.

Regarding the presentation of results, I believe it could be greatly improved by addition of at least a graphical resource (e.g. barplot). Only two tables are shown and some additional data is directly presented in the text. The total number of captures is shown in Table 2 and the distribution of species per year and location is presented in Table 3. However, only partial results on trap type performance for every insect species is presented in results and discussion. A deeper analysis and representation of the existing data would, in my opinion, improve the readability and reinforce the relevance of the study. Discussion of differential sampling effort and airport size on detection sensitivity is specially recommended.

As for the discussion, some results are presented for the first time in this section, which should be introduced earlier. I believe that a more systematic analysis of the results might provide more sound or even additional conclusions and suggestions.

The discussions need to be fully supported and properly analyzed in relation to the presented data. For instance: “The main reason for the multiple EMS captures in The Netherlands at Schiphol can be attributed to the large total volume of passengers and cargo transported.” Although such a statement might be true, the discussion should also include the aspect of comparative sampling effort (maybe adjusted for airport size). 

Dear reviewer, thanks for your comments on the submitted manuscript. In this document we provide specific answers to the comments. Regarding the general comments, we believe that in the revised manuscript we improved the definition of the specific sample locations and methodology, the analysis of the results, and the discussion on the differences on sampling effort and airport size on the studied airports. We acknowledge the suggestion of the addition of a graphical source. However, our sampling results does not produce a graph that will provide better quality than the provided tables.

Specific comments:

Lines 77-94. This section initially leads the reader to believe that all mentioned airports are included in this study, while data for only three airports is provided in the results section. Such contextualization on current European airport sampling should be stated in the introduction, not in M&M.

Information of the complete list of airports in UK, Belgium and The Netherlands has been included in the introduction as suggested (lines 72-79). Only the airports included in the study are now included in the M&M section (lines 85-86).

Line 117. “Standard oviposition traps (OT)” instead of “Standard oviposition (OT) traps”.

Corrected, line 122

Line 144. I believe the registered trademark is Mosquito Magnet, not the company: “Mosquito Magnet® Independence (MM trap) (WoodstreamTM Co., Lititz, USA)”. Please check.

Thanks for this comment. This has been changed in revised manuscript, line 149

Line 153: Suggestion: “OT were controlled every four weeks. Hay infusion was used as a lure in 2017”

Changed as suggested, line 158

Line 153: Is there an expected decrease in attractiveness due to lack of attractant in years other than 2017?

We don’t expect a significant decrease in attractiveness due to the lack of hay as attractant. The use of hay in the traps is recommended in the ECDC guidelines for surveillance of invasive mosquitoes only if several breeding sites are available. In Liege, breeding sites are absent in the surveyed site and that was the main reason to stop using hay.

Line 188: I’d suggest a new paragraph for this sentence.

Changed as suggested, line 194

Line 189: Suggestion: “sequencing of three different DNA regions (COI, Nad4 and ITS2)”

Changed as suggested, line 195

Lines 190-199: Cycling conditions are made explicit for only one DNA region. Please consider adding a table with PCR region, primers, cycling conditions and references. Also, try to clearly discern when talking about Amplification and Sequencing.

As suggested by the reviewer we have added two tables (in supplementary material Table S1, and Table S2) with PCR region, primers, cycling conditions and references. In the main manuscript this information can be found at lines 196-199

Line 262: Both Ae. aegypti an Ae. albopictus captured only in BG-M traps. Reporting the detail of captures depending on trap type would help. It might also lead to discussion on trap type pros and cons and possible specificities.

Detail of the Ae. albopictus and Ae. aegypti captures depending on the trap type has been added in lines 267-273.

Line 279: Do you think the trap type is relevant?

Line 279 refers to the manually collection of Mansonia specimens, and it is not related to trapping. About the question of the relevancy of the trap type, we think that indeed, the trap type can influence the captured species at airports. Lures used in BG-mosquitaire traps are designed to capture invasive Aedes species Ae. albopictus and Ae. aegypti. Other mosquito species are also attracted to these trap types. BG-GAT type or OT, are specifically designed to detect container breeding exotic Aedes species such as Ae. albopictus and Ae. aegypti. In our case, the used BG-traps seems to be efficient for Aedes species, but without additional CO2 these traps could be less attractive for other blood-seeking mosquitoes of other genus such as Anopheles, Culex, etc. The use of MM-traps and the capture of An. pharoensis demonstrates that CO2 produced by the trap can attract this Anopheles species. Addition of CO2 is recommended for increase the spectrum of mosquito species attracted by the traps, but the use of CO2 tanks (for BG-M) or propane tanks (for MM-trap) inside airports its difficult due to security and logistic reasons.

Line 288: I’d suggest to use colon (were in line: four specimens...)

Changed as suggested, line 299

Line 295: Missing Accession Codes.

GenBank accession number are now added in lines 306-307

Lines 322-324: Besides published data, the findings of the present paper in UK airport should be more clearly stated.

This sentence has been adapted, lines 334-336

Line 333-335: No temporal data is provided. When where they captured? During winter only indoor traps (all of them in cargo areas) were deployed, so this information could be relevant, as traps out of baggage area seem to be deployed only during summer.

As suggested, temporal data has been included as text in the results section indicating the number of Ae. aegypti specimens captured indoors and outdoors, lines 262-267.

Lines 347-350: This information could be detailed in M&M, as it does not seem to add to the discussion.

As suggested, this information has been detailed in M&M, lines 109-113.

Line 355: please avoid repetition: “Since these traps are especially designed for trapping Aedes species, they could…”

Changed as suggested, line 363

Line 357-359: The fact that a trap is able to capture some insects seems to be presented here as demonstrative of its efficiency. However, insect capture by itself is not proof that the trap is not suboptimal (i.e. less efficient than a more specific trap type). Since evidence is not provided in that sense, please reword the sentence.

The sentence has been reworded as suggested, lines 365-367

Line 363-364: I believe that these results have not been mentioned in the Results section.

The information on the spread of the findings of Ae. aegypti over the year has been moved to the Results section, lines 262-267.

Line 366: In Table 1, 2018 shows a sample number increase in Schiphol of about 70% (1441 vs 853). I would not say the sampling effort has not changed.

The sentence has been reworded as suggested, lines 372-376

Line 367: The placement of extra traps is not mentioned in M&M. Please do so.

The placement of extra traps following a EMS finding is mentioned in M&M, lines 135-138.

Line 369: It is not clear how the trap type is relevant for this conclusion. Please clarify or remove the mention to trap type.

As suggested BG-M trap has been removed as it is not relevant for the conclusion. Lines 374-376

Line 370: Please clarify if the reference of ‘These traps’ is meant for both trap types (OT and BG-GAT) or only OT.

Corrected as suggested, lines 376-377

Lines 376: What is considered a 'good sampling frequency'? Sampling frequency is not mentioned in M&M. Justify the reasons that support the recommendation for biweekly sampling.

The sampling frequency is mentioned in M&M for each trap type. As suggested we justify the main reason (latitude) that support the recommendation for biweekly sampling at the countries included in the study. Lines 382-387.

Line 383: This information on Aedes first report might be better in the introduction.

This sentence has been removed from the manuscript.

Line 428: it does not make much sense to refer to the development stage of a species in a molecular identification result. Please reword (e.g: The immature specimen collected…)

This issue has been corrected in the revised version of the manuscript, line 434-435

Lines 477-478: I believe the sampling effort should also be introduced into discussion here.

As suggested, relative sampling effort has been introduced in this paragraph (lines 484-486) indicating a larger effort in Schiphol airport, compared to London and Liege airports.

Table 1. Please check formatting of column labels.

Formatting table corrected

Table 3. Bold type seems to indicate non-indigenous species but, if so, it should be clearly explained in the table caption. Also, Anopheles crucians is not in bold for 2016 data point (is it a mistake?). Please check format: the term ‘indigenous’ is not consistently italicized.

As suggested, bold type indicating non-indigenous species is now indicated in the table title, results on Anopheles crucians are now in bold type, and italicization of “indigenous” term has been removed.

Reviewer 2 Report

This study presented data on trap captures of the exotic mosquitoes at several international airports in Europe and demonstrates the route or pathway of accidental transports or introduction of several nonnative mosquito species.  The manuscript is fairly well written, but may be approved by additional editing.  My major suggestion is to present some descriptive statistics (e.g., mean and SE) on traps captures at different airports.

Minor comments:

Throughout the manuscript, when talking about the "introduction" of mosquitoes, please add "accidental" or "inadvertent" before the introduction.

L 22 - 24.  This sentence needs revision - "...we.....coordinated invasive mosquito project", it's grammatically not correct.

Author Response

This study presented data on trap captures of the exotic mosquitoes at several international airports in Europe and demonstrates the route or pathway of accidental transports or introduction of several nonnative mosquito species.  The manuscript is fairly well written, but may be approved by additional editing.  My major suggestion is to present some descriptive statistics (e.g., mean and SE) on traps captures at different airports.

Thanks for the comments and suggestions. In the revised version of the manuscript we provide the information on EMS trap captures at different airports. Since the manuscript is focused on EMS detections, we provide in the Table 2 the mean and standard error of samples containing EMS of each trap type.

Minor comments:

Throughout the manuscript, when talking about the "introduction" of mosquitoes, please add "accidental" or "inadvertent" before the introduction.

As suggested, "accidental" or "inadvertent"  has been added before the “introduction” in the revised version of the manuscript, in lines 18, 26, 28, 52, 356, 359, 393, and 505 for example.

L 22 - 24.  This sentence needs revision - "...we.....coordinated invasive mosquito project", it's grammatically not correct.

Sentence has been revised, lines 23-24

Reviewer 3 Report

“Detection of exotic mosquito species (Diptera: Culicidae) at international airports in Europe” by Ibáñez-Justicia et al. presents findings of surveillance efforts for non-native mosquito species at international airports in three European countries carried out by the Centre for Monitoring of Vectors (CMV) of the Netherlands, by the MEMO project (Monitoring of Exotic MOsquitoes) in Belgium, and the Public Health England coordinated invasive mosquito surveillance project.  Native and exotic mosquito species collected by a variety of trap types were identified. The morphological identifications of the exotic species were validated using molecular tools. The morphological identifications of Aedes aegypti, Ae. albopictus, Anopheles crucians, An. subpictus and An. pharoensis were confirmed by molecular methods. However, the morphology-based identification of the Mansonia specimens could not be validated using DNA sequences. The findings of the study demonstrate the aircraft-mediated transport of mosquitoes into Europe from a wide geographical range and the value of such surveillance programs. The study also demonstrates the pressing need for molecular characterization of Mansonia species.

The paper is well written and neighbor joining trees are robust.  The manuscript includes an extensive bibliography of the relevant publications on the diverse topics covered in the paper.

There are several issues with the typesetting that are problematic.  (1) The headings in the tables spill over in adjacent lines of text.  The margins of the text boxes in the table should be set at widths to ameliorate this issue.  (2) The font of the text of the manuscript and the References is not consistent.  (3) The positioning of the figure leaves a large gap of empty space.  This needs to be corrected.

Minor comments:

Line 144: (…Lititz, PA, USA…)

Table 3: Why is Schiphol in bold font?

Line 262: Close up text and comma.

Line 320: Correct “5’ ” to “5.”

Lines 325, 329, 394, 488, 495, 501, 502: “…aircraft…”  Aircraft is an invariant noun; “aircrafts” is is not recognized as the pleural form.

Line 362 Correct the spelling to be consistent: tire vs (line 385) tyres

Lines 613-615: Correct the capitalization in the citation.

Lines 734, 800: Italicize Aedes albopictus.

Lines 822, 825: Italicize Anopheles crucians.

Lines 692, 835, 850: Italicize Anopheles.

Author Response

#Reviewer 3

“Detection of exotic mosquito species (Diptera: Culicidae) at international airports in Europe” by Ibáñez-Justicia et al. presents findings of surveillance efforts for non-native mosquito species at international airports in three European countries carried out by the Centre for Monitoring of Vectors (CMV) of the Netherlands, by the MEMO project (Monitoring of Exotic MOsquitoes) in Belgium, and the Public Health England coordinated invasive mosquito surveillance project.  Native and exotic mosquito species collected by a variety of trap types were identified. The morphological identifications of the exotic species were validated using molecular tools. The morphological identifications of Aedes aegypti, Ae. albopictus, Anopheles crucians, An. subpictus and An. pharoensis were confirmed by molecular methods. However, the morphology-based identification of the Mansonia specimens could not be validated using DNA sequences. The findings of the study demonstrate the aircraft-mediated transport of mosquitoes into Europe from a wide geographical range and the value of such surveillance programs. The study also demonstrates the pressing need for molecular characterization of Mansonia species.

The paper is well written and neighbor joining trees are robust.  The manuscript includes an extensive bibliography of the relevant publications on the diverse topics covered in the paper.

There are several issues with the typesetting that are problematic.  (1) The headings in the tables spill over in adjacent lines of text.  The margins of the text boxes in the table should be set at widths to ameliorate this issue.  (2) The font of the text of the manuscript and the References is not consistent.  (3) The positioning of the figure leaves a large gap of empty space.  This needs to be corrected.

Thanks for the comments and suggestions. In the revised version of the manuscript the table headings have been corrected to solve the mentioned issue. The font of the text and references will be consistent after receiving the final proof from the editor, and the positioning of the figure will be corrected by the editorial service after receiving the final proof from the editor.

Minor comments:

Line 144: (…Lititz, PA, USA…)

Corrected as suggested, line 149

Table 3: Why is Schiphol in bold font?

Corrected as suggested in Table 3.

Line 262: Close up text and comma.

Corrected as suggested, line 272

Line 320: Correct “5’ ” to “5.”

Corrected as suggested, line 332

Lines 325, 329, 394, 488, 495, 501, 502: “…aircraft…”  Aircraft is an invariant noun; “aircrafts” it is not recognized as the pleural form.

Corrected as suggested in the mentioned lines. The word “aircrafts” is not used in the revised version of the manuscript.

Line 362 Correct the spelling to be consistent: tire vs (line 385) tyres

Corrected as suggested, line 370

Lines 613-615: Correct the capitalization in the citation.

Corrected as suggested, line 613-314

Lines 734, 800: Italicize Aedes albopictus.

Corrected as suggested, lines 722, 772-773

Lines 822, 825: Italicize Anopheles crucians.

Corrected as suggested, lines 785, 794

Lines 692, 835, 850: Italicize Anopheles.

Corrected as suggested, lines 684, 803 815

Round 2

Reviewer 1 Report

Dear authors,

I believe all the concerns and suggestions I raised regarding the first version of the manuscript have been adequately revised or justified.

Let me congratulate you for the work which I think will be a fine addition to the knowledge on the risks of invasive mosquito species introduction.

My best regards